# The Association of Cerebral Autoregulation Dysfunction and Postoperative Memory Impairment in Cardiac Surgery Patients

**DOI:** 10.3390/medicina60081337

**Published:** 2024-08-17

**Authors:** Greta Kasputytė, Birutė Kumpaitienė, Milda Švagždienė, Judita Andrejaitienė, Mindaugas Gailiušas, Edmundas Širvinskas, Arūnas Gelmanas, Yasin Hamarat, Edvinas Chaleckas, Vilma Putnynaitė, Laimonas Bartušis, Rolandas Žakelis, Vytautas Petkus, Arminas Ragauskas, Tadas Lenkutis

**Affiliations:** 1Institute of Cardiology, Medical Academy, Lithuanian University of Health Sciences, 50162 Kaunas, Lithuania; 2Department of Disaster Medicine, Medical Academy, Lithuanian University of Health Sciences, 44307 Kaunas, Lithuania; 3Department of Cardiac, Thoracic and Vascular Surgery, Lithuanian University of Health Sciences, 50161 Kaunas, Lithuania; 4Department of Anesthesiology, Medical Academy, Lithuanian University of Health Sciences, 44307 Kaunas, Lithuania; 5Health Telematics Science Institute, Kaunas University of Technology, 51423 Kaunas, Lithuanialaimonas.bartusis@ktu.lt (L.B.);

**Keywords:** cerebral autoregulation, memory impairment, cardiac surgery

## Abstract

*Background and Objectives*: Cardiac surgery is associated with various durations of cerebral autoregulation (CA) impairment and can significantly impact cognitive function. Cognitive functions such as memory, psychomotor speed, and attention are significantly impacted after cardiac surgery, necessitating prioritization of these areas in cognitive function tests. There is a lack of research connecting cerebral autoregulation impairment to specific cognitive function domains after cardiac surgery. This study aimed to determine if impaired cerebral autoregulation is associated with postoperative memory impairment and to test the hypothesis that the duration of this impairment affects the development of postoperative memory issues. *Materials and Methods*: A prospective study was conducted in 2021–2023. After approval of the Ethics Committee and with patient’s written consent, 83 adult patients undergoing elective on-pump coronary artery bypass graft (CABG) surgery were enrolled. All patients were assessed for cognitive function 1 day before surgery using the Mini-Mental state examination (MMSE-2) test as a screening tool and the Hopkins Verbal Learning Test-Revised (HVLT-R) to assess memory specifically. To diagnose possible memory impairment (IM), all patients underwent a repeat assessment of cognitive function on the 7th–10th postoperative day. Cerebral autoregulation monitoring using transcranial Doppler was performed. Cerebral autoregulation status index (Mx) was recorded using Intensive Care Brain Monitoring System software, 9.1.5.23 (Cambridge, UK). *Results*: According to our research, the incidence of postoperative memory impairment is 30.1%. Temporary cerebral autoregulation impairment occurs in all patients undergoing elective in-pump CABG surgery. The duration of the single longest CA impairment event in seconds (LCAI) and the LCAI dose were higher in patients with postoperative memory impairment, *p* = 0.006 and *p* < 0.007, respectively. *Conclusions:* Cerebral autoregulation impairment is important in developing memory loss after cardiac surgery. The duration and dose of the LCAI event are predictive of postoperative memory impairment.

## 1. Introduction

A delayed recovery of neurocognitive function (dNCR) affects a substantial portion of patients, ranging from 20 to 50% after cardiac surgery [1]. This delay in recovery impacts patients’ memory, learning ability, motor skills, and even their self-care ability. These impairments lead to longer hospital stays, increased patient care costs, and, most importantly, a significant decline in patients’ quality of life [2].

In addition to established risk factors such as older age, lower education level, cognitive impairment, diabetes, surgery, anesthesia and cardiopulmonary bypass (CPB) duration and other factors, impaired individual cerebral autoregulation (CA) significantly affects patients’ cognitive functions during cardiac surgery. Cognitive functions such as memory, psychomotor speed, and attention are significantly impaired after cardiac surgery, necessitating prioritization of these areas in cognitive function tests [3,4]. The previous literature usually presents findings for the general impairment of cognitive functions after cardiac surgery rather than focusing on specific domains. Therefore, we conducted our research specifically on memory impairment. Furthermore, in current clinical studies, researchers choose different cognitive function tests based on factors such as performance duration, validity in a particular country, financial considerations, and personal opinions about which tests are best for assessing postoperative cognitive functional disorders after cardiac surgery [4,5].

Additionally, there is a lack of research connecting CA dysfunction to specific cognitive functional domains after cardiac surgery. The association of cerebral autoregulation with the development of postoperative memory impairment in Lithuania has not been widely studied.

This study aimed to determine if impaired cerebral autoregulation is associated with postoperative memory impairment and to test the hypothesis that the duration of this impairment affects the development of postoperative memory issues.

## 2. Materials and Methods

This study was approved by the Kaunas Regional Biomedical Research Ethics Committee (No. P1-BE-2-64/2021, date: 15 December 2021), and participants provided written informed consent according to the Declaration of Helsinki. This study enrolled 83 patients undergoing elective on–pump coronary artery bypass graft (CABG) surgery with CPB at the Department of Cardiothoracic and Vascular Surgery in the Hospital of Lithuanian University of Health Sciences Kaunas Clinics. All patients signed the written consent form and met the inclusion criteria: elective heart surgery with CPB, no anamnesis of neurocognitive disease, not using agents affecting the central nervous system (CNS) (such as benzodiazepines or antidepressant medications), and carotid artery atherosclerosis ≤ 50%. Patients with anamnesis of neurological disorders and uncontrolled diabetes (HbA1c ≥ 7%) were excluded from the study.

All patients underwent pre-surgery cognitive function assessment using the Mini-Mental State Examination (MMSE-2) as a screening tool and the Hopkins Verbal Learning Test—Revised (HVLT-R) to evaluate memory specifically.

All patients underwent the same surgical, anesthetic, and cardiopulmonary bypass circulation techniques.

For premedication, 1–2.5 mg of lorazepam and a half dose of the patient’s regular daily dose of metoprolol were administered. General anesthesia was induced according to the standard hospital protocol after preoxygenation by face mask with 80% oxygen with fentanyl 1–2 μg/kg, propofol 2 mg/kg, and rocuronium 0.6 mg/kg intravenously. Anesthesia was maintained with sevoflurane according to the bispectral index (BIS 40–60) and fentanyl (10–12 μg/kg) for analgesia and muscle relaxants as needed. The lungs were ventilated with a mixture of 50% oxygen to maintain normocapnia (ETCO2 35–35 mmHg). Middle sternotomy section and standard surgical technique for anastomosis formation were performed for all of the patients. For systemic heparinization, an initial dose of 3 mg/kg of unfractionated heparin was given to achieve an active clotting time (ACT) ≥ 400 s. The CPB circuit was primed with 1500 mL of Ringers’ acetate crystalloid solution and 1000 IU of unfractionated heparin. Pump flow was maintained at a 2.2–2.4 L/min/m^2^ cardiac index. The temperature during CPB was 35–36 °C. For myocardial protection, a cold St. Thomas cardioplegic solution was used. The initial dose of cardioplegia was 1000 m, followed by 500 mL after every 35–40 min. After weaning from CPB, anticoagulation was reversed by a dose of 1.2 mg/kg of protamine sulfate intravenously. Adequate reversal was controlled with an ACT time of 120–140 s. Tranexamic acid was administered routinely thrice during surgery at a quantity of 1 g according to a clinical protocol. 

The cerebral autoregulation was monitored during surgery using the ultrasonic robotic two–channel transcranial Doppler (TCD). Monitoring was performed using a special head frame with a pair of ultrasonic transducers positioned on opposite sides of the patient’s temporal bones, which was used to transmit and receive an ultrasound signal and to monitor real-time middle cerebral arteries’ blood flow velocities (vMCA) in both hemispheres of the patient’s brain. Arterial blood pressure (ABP) and the cerebral autoregulation status, namely the mean flow index (Mx), were recorded using Intensive Care Brain Monitoring System (ICM+) software, 9.1.5.23 (Cambridge, UK). ABP and vMCA signals were filtered with a 0.1 Hz cutoff frequency lowpass Butterworth filter for Mx. Pearson’s correlation was calculated in a 30 s window due to the pulse length. The threshold of CA intact is Mx <0.4, and for impaired CA, Mx is >0.4. In the analysis, we calculated the duration of the single longest cerebral autoregulation impairment (LCAI) event in seconds and its dose. The dose was calculated as an area under the curve (AUC) for each episode when Mx > 0.4. In addition, the total duration and total dose of all CA impairment events were calculated to compare their influence with the influence of the single longest CA impairment on memory impairment.

To diagnose possible memory impairment, all patients underwent a repeat assessment of cognitive function on the 7th–10th postoperative day. The cut-off 36 (T score of recognition discriminant index) for HVLT—R test was chosen to determine postoperative memory impairment.

Statistical analysis was performed using IBM Statistical Package software, version 29.0 (IBM SPSS). The normality of data was assessed with the Kolmogorov–Smirnov or Shapiro–Wilks tests. Qualitative variables were compared using the Pearson chi-square test. Quantitative characteristics that did not meet the definition of normality were compared using the Mann–Whitney U criterion (data are presented as median Q1–Q3, minimum–maximum values), and differences between pre- and postoperative points of the HVLT-R test were calculated using Wilcoxon test. The differences were considered statistically significant if the *p*-value was lower than the significance level of 0.05.

## 3. Results

A total of 108 patients were screened for eligibility, and the 83 included patients were subjected to further data analysis (Figure 1).

This study analyzed patients’ memory pre- and post-surgery. Our research has shown that the rate of postoperative memory impairment is 30.1%. After undergoing the HVLT-R memory evaluation, patients were divided into two groups: non-impaired memory (non-IM) and impaired memory (IM). The patients’ demographic data are shown in Table 1.

The results showed that despite an overall postoperative memory impairment rate of 30.1% (based on a set cut-off point of 36), 49.4% of patients showed a lower recognition discriminant index (T score) after surgery than before (Table 2).

Our data have shown that temporary CA impairment occurs in all patients undergoing elective on-pump CABG surgery. There were statistically significant results in patients with postoperative memory impairment. CA monitoring data are shown in Table 3.

Our study found that patients who experienced postoperative memory impairment had longer durations of cerebral impairment events in seconds, *p* = 0.006, and higher event doses, *p* < 0.007. Figure 2 illustrates an example of one patient’s single longest CA impairment event in seconds and the dose of this event.

## 4. Discussion

This study aimed to determine the association between CA impairment and the incidence of postoperative memory impairment in cardiac surgery patients. During the study, we monitored the patients’ CA status from their arrival in the operating room while they were conscious until the end of the surgery.

In our study, it was discovered that all patients had varying durations of CA impairment. The shortest episode lasted 30 s, while the longest was 2692 s. We observed statistically significant associations between the duration and the dose of the single longest CA impairment event in seconds and memory impairment after surgery, *p* = 0.006 and *p* < 0.007, respectively. This is consistent with the findings of other researchers, who have identified CA impairment as a contributing factor to cognitive function decline following cardiac surgery [6,7,8]. Most studies have only measured CA during CPB or other specific moments of the intraoperative period, potentially missing instances of CA disturbances [7].

We also found that the duration of the surgery was associated with postoperative memory impairment. Surgery was longer in patients diagnosed with memory impairment after surgery, *p* = 0.01. These findings are supported by a systematic review by Graves D et al. In their meta-analysis, the duration of the surgery is specified as a separate intraoperative risk factor for post-CABG cognitive decline [9]. Despite this data, some research has produced opposite results. Sun Y et al. did not find a relationship between the duration of surgery and the occurrence of cognitive impairment in CABG patients [10].

It is worth noting that the monitoring methods used for CA status classification can impact the results. For example, a study by Ono et al. in 2012 found that the frequency of CA impairment during cardiac surgery was 20% when monitored with TCD, whereas, in 2013, the same authors found it was only 11% when monitored using near-infrared spectroscopy (NIRS) [11,12]. We monitored CA status by measuring the blood flow velocity in the MCA using transcranial Doppler. TCD is a reliable monitoring method but is not commonly used in routine clinical practice due to the specific skills and accuracy required [13]. In our study, a dedicated team of experts placed the head helmet with TCD transducers and continuously monitored CA throughout the surgery. We also believe that in the future, CA monitoring could be used as an additional tool during surgery. This could provide anesthesiologists with information about possible CA disorders and help identify remediable causes to reduce the risk of postoperative neurological complications, including memory impairment, after cardiac surgery.

A review by Tony Wu et al. suggests that evaluating memory, attention, and psychomotor speed should be the focus of postoperative cognitive assessment after cardiac surgery [4]. We performed the HVLT-R test for all patients, specifically designed to assess one of the main cognitive functions (memory). The Special Psychology Laboratory of Vilnius University adapted and standardized this test in Lithuania. The HVLT-R test consists of three recall learning trials, a delayed recall trial, and a recognition task. Several aspects of memory are assessed: reproduction, recognition, differentiation of memorized and new material, and the learning process. The Recognition Discriminant Index is the most important tool for researchers and clinicians. It measures the difference between the number of correct responses and the number of false positives. As far as we know, this is the only test whose results have been evaluated in relation to a certain age group of patients. Therefore, it provides a much more accurate diagnosis of memory disorders [14].

Our study found that a patient-specific, single-longest CA impairment event and the dose of this event recorded for individual patients intraoperatively were statistically significantly associated with developing one of the cognitive domains: memory impairment after on-pump CABG surgery.

Advancing age and lower levels of education are the two key factors that significantly affect postoperative memory loss [15]. Interestingly, our study showed that the patients with the single longest critical CA impairment event and memory impairment were younger but had a shorter education duration than those without impaired memory. These findings align with a 2017 meta-analysis by Feinkohl I et al., which suggests that education level is crucial in determining a patient’s cognitive reserve. Cognitive reserve is like a brain mechanism that supports specific compensatory thinking, creating a new neural network and compensating for certain damage that has occurred, in this case, CA dysfunction during cardiac surgery. A longer duration of education is linked to a lower risk of delayed recovery of neurocognitive functions after various operations [16]. Our data back up such findings.

Several other factors can significantly impact the decline of cognitive abilities following cardiac surgery. These include a history of mental health issues, cerebrovascular disease, pre-existing cognitive impairment, the type of surgery, blood transfusions during surgery, the use of risperidone, postoperative atrial fibrillation, and prolonged mechanical ventilation [17]. None of the patients in our study had a preoperative diagnosis of psychiatric or other neurological disorders. All patients underwent elective CABG surgery, and before that, a Mini-Mental state examination test was performed to screen patients for preoperative cognitive impairment assessment.

While our study was informative, it is important to recognize that it had some limitations. The sample size was relatively small. A larger study is necessary to make meaningful progress in this area. Additionally, we did not evaluate long-term mild or major neurocognitive disorders, a critical aspect of the topic that deserves further attention. Despite these limitations, the significance of this topic cannot be understated, and we believe this could be the subject of further research.

## 5. Conclusions

In conclusion, our findings support the hypothesis that cerebral autoregulation impairment is important in developing memory loss after cardiac surgery. The duration and the dose of the single longest cerebral autoregulation impairment event are predictive factors for delayed recovery of neurocognitive functions, particularly memory impairment.

## Figures and Tables

**Figure 1 medicina-60-01337-f001:**
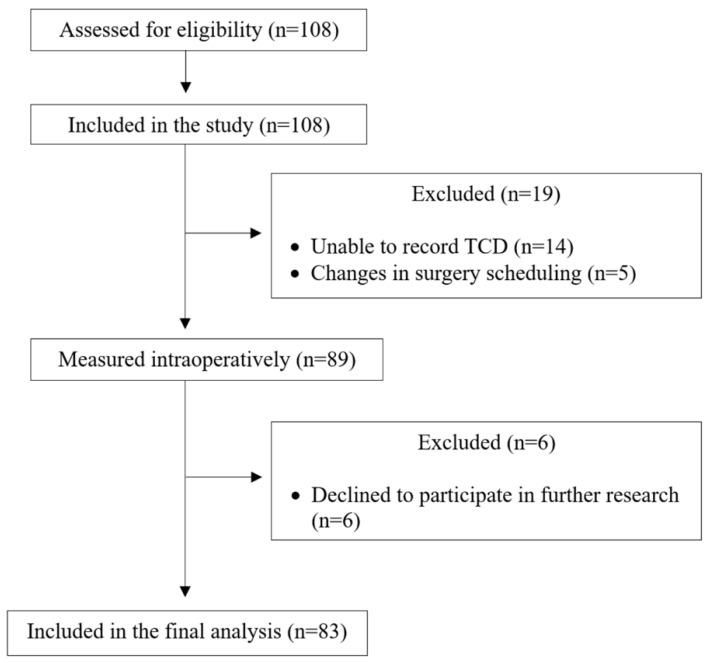
Flow chart of the study.

**Figure 2 medicina-60-01337-f002:**
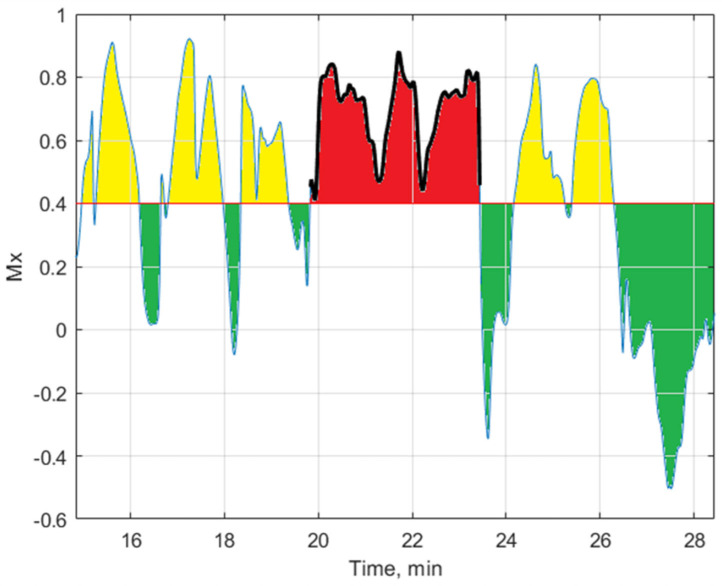
Cerebral blood flow autoregulation status identification index Mx (t). Critical threshold Mx = 0.4. Green—intact CA, yellow—short impairment of CA, which the patient’s brain tolerates. Red—single longest CA impairment event in seconds, whose duration is 217 s. The dose of the single longest CA impairment event is the red area under the single longest CA event curve.

**Table 1 medicina-60-01337-t001:** Patient’s demographic data according to HVLT-R test results.

Variables	Non-IM Group (n = 58)	IM Group (n = 25)	*p*-Value
Age (years)			
Median (min–max) Quartiles (Q1–Q3)	70.5 (55–85) (64–75)	58 (55–79) (55.0–63.5)	<0.001 *
Sex			
Male, n (%) Female, n (%)	39 (67.2) 19 (32.8)	14 (82.4) 3 (17.6)	0.229
Years of education			
Median (min–max) Quartiles (Q1–Q3)	13 (7–25) (11–16.5)	11.0 (4–15) (11–11.5)	<0.001 *
Duration of cardiopulmonary bypass (min)			
Median (min–max) Quartiles (Q1–Q3)	82 (52–125) (72–94)	84 (56–166) (36.5–54.5)	0.462
Duration of surgery (min)			
Median (min–max) Quartiles (Q1–Q3)	180 (140–240) (168.75–196.25)	210 (125–330) (180–240)	0.010 *
Duration of aortic cross-clamp (min)			
Median (min–max) Quartiles (Q1–Q3)	40 (23–65) (33–46.25)	47 (29–116) (36.5–54.5)	0.096

Abbreviations: HVLT-R, Hopkins Verbal Learning Test-Revised; IM, impaired memory after surgery; min, minutes. Data are presented as the median (min, max, Q1, Q3) or a proportion %, as appropriate. * *p*-value—Mann-Whitney test, results are statistically significant.

**Table 2 medicina-60-01337-t002:** Patient’s memory assessment according to HVLT-R test results.

Variables	Pre-Surgery	Post-Surgery	*p*-Value
T score of recognition discrimination index			
Median (min-max) Quartiles (Q1–Q3)	53 (20–65) (45–59)	48 (20–68) (37–53)	<0.001 *

Data are presented as the median (min, max, Q1, Q3). * *p*-value—Wilcoxon test, results are statistically significant.

**Table 3 medicina-60-01337-t003:** Cerebral autoregulation parameters in patients during surgery.

HVLT-R Test			
Variables	Non-IM Group	IM Group	*p*-Value
Duration of single longest cerebral autoregulation impairment event in seconds, Mx			
Median min–max Q1–Q3	447.5 30–1610 234.8–837.3	794.5 337–2692 517.5–1458.3	0.006 *
Duration of total cerebral autoregulation impairment events in seconds, Mx			
Median min–max Q1–Q3	4145.5 860–6372 3593.8–4730	4556 2905–9573 3903.8–5989.3	0.139
Dose of single longest cerebral autoregulation impairment event in seconds, Mx			
Median min–max Q1–Q3	422.95 21.6–1542.6 204.6–789.5	756,54 303.58–2607.8 470.6–1383.9	<0.007 *
Dose of total cerebral autoregulation impairment events in seconds, Mx			
Median min–max Q1–Q3	3662.3 509.9–5622.7 3162.9–4114	3877.8 2724–8605.7 3373.3–5503.3	0.073

Abbreviations: HVLT-R, Hopkins Verbal Learning Test-Revised; IM, impaired memory after surgery; * *p*-value—Mann-Whitney test, results are statistically significant.

## Data Availability

The data presented in this study are available upon request from the corresponding author.

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
