# Peer review of "The Association of Cerebral Autoregulation Dysfunction and Postoperative Memory Impairment in Cardiac Surgery Patients"

_medicina, 2024, doi:10.3390/medicina60081337_

Round 1

Reviewer 1 Report

Comments and Suggestions for Authors

This study aimed to determine if impaired cerebral autoregulation is associated with postoperative memory impairment and to test the hypothesis that the duration of this impairment affects the development of postoperative memory issues.

 A few section-wise suggestions/changes are required to further improve the manuscript.

Abstract:

1. Please remove the abbreviations like CABG, MMSE-2, HVLT-R, and IM, as they are not used again in the abstract.

2. In line 15, remove the statement " prioritization of these areas in cognitive function tests" to avoid duplication.

2. In line 34, the authors mentioned that “reassessment of cognitive function was made on 7-10 days”. Please clear this statement whether the post-operative assessment was made repeatedly on 7, 8, 9, and 10 days or it was done once on these days depending upon the availability/accessibility of the patient.

3. Line 37, " memory impairment is 30.1%" may be written as “memory impairment was 30.1%.”

4. Line 39-40, " Duration of single longest CA impairment event in seconds (LCAI) and LCAI dose were higher in patients with postoperative memory impairment, p=0.006 and p<0.007, respectively", this statement can be rephrased as “Duration and dose of single longest CA impairment event in seconds were higher (P<0.0.5) in patients with postoperative memory impairment”.

Introduction:

1. Please remove the abbreviations dNCR, and CPB, as they are not used again in the introduction.

2. In lines 66-67, "The association of cerebral autoregulation with the development of postoperative memory impairment in Lithuania has not been widely studied". Please explain this statement if this study is aimed to assess the association of cerebral autoregulation with the development of postoperative memory impairments specifically in patients residing in Lithuania, then the name of the country might be added to the title.

3. Overall, the introduction requires thorough revision to avoid the repetition of statements.

Materials and Methods:

1. Please add the duration of collecting the data of patients which is missing in this section.

2. The protocol of CABG used in this study is novel? If not, then please make it brief by explaining the steps that are taken in addition to the existing protocol of CABG.

3. Please follow comment number 1 for materials and methods too.

Results

1. It is suggested that the flow chart might be shifted to the material and methods section.

2. Lines 153-154, “After undergoing 153 the HVLT-R memory evaluation, patients were divided into two groups: non-impaired 154 memory (non - IM) and impaired memory (IM).” Please also add/explain this in the material and method section.

3. Lines 161-163, “The results showed that despite an overall postoperative memory impairment rate of 30.1% (based on a set cut-off point of 36), 49.4% of patients showed a lower recognition discriminant index (T score) after surgery than before (Table 2)”. Please check whether the percentages mentioned in the text are present in Table 2.

4. Line 167-169, “Our data have shown that temporary CA impairment occurs in all patients undergoing elective on-pump CABG surgery. There were statistically significant results in patients with postoperative memory impairment. CA monitoring data are shown in Table 3.” Please write the results specifically by interpreting each variable of Table 3 whether significant or nonsignificant instead of writing the general statements.

Discussion:

1. In line 191, please replace "p<0.007" with "p=0.007".

2. Please follow the same sequence of discussing the variables as mentioned in the results.

Reviewer 2 Report

Comments and Suggestions for Authors

The article is devoted to an urgent problem - assessing the likelihood of developing postoperative cognitive dysfunction in case of dysfunction of the autoregulation of cerebral blood flow to assess the prognosis and diagnosis of this dysfunction in adult cardiac surgery. Notes on the article.

I think the example in Figure 1 is superfluous; if this technology is official, then there is no need for examples. Were the patients in the group with and without cognitive dysfunction comparable in terms of education, level and nature of work and main work activity? The most important questions are: what was the cause of the autoregulation disorder in a particular clinic, provided that the patients in the groups were comparable in terms of the type of anesthesia, the nature and technology of artificial blood circulation, the time of aortic clamping and the number of coronary anastomoses performed?

Reviewer 3 Report

Comments and Suggestions for Authors

Many thanks for asking me to review this interesting manuscript

This is a prospective nonrandomized study (2021 – 2023). 83 adult patients underwent elective on-pump coronary artery bypass graft (CABG) surgery and assessed for cognitive functions using Mini-Mental state examination (MMSE-2) test as a screening tool and Hopkins Verbal Learning Test-Revised (HVLT-R) to assess memory specifically.

Cerebral autoregulation was monitored during surgery using transcranial Doppler with cerebral autoregulation status index.

Authors found 30% had postoperative memory impairment. Cerebral autoregulation impairment was seen in all. Duration of single longest CA impairment event (LCAI) and LCAI dose were higher in patients with postoperative memory impairment, p=0.006 and p<0.007, respectively.

The study is well conducted and described well in the methods section. The results of the study are well elucidated with appropriate tables. The discussion is well written and organized. Conclusions of the study are supported by the results.

The article is well written and researched with appropriate bibliography.

I would recommend publication.

Please consider the following minor comments and suggestions

Line 23. Please correct repetition ‘…prioritization of these areas in cognitive function tests.’

Line 103. Please correct 1000m to 1000ml. ‘….initial dose of cardioplegia was 1000 m’

Round 2

Reviewer 1 Report

Comments and Suggestions for Authors I went through the file attached in the email. The changes made are satisfactory.